# High NIR Reflectance and Photocatalytic Ceramic Pigments Based on M-Doped Clinobisvanite BiVO_4_ (M = Ca, Cr) from Gels

**DOI:** 10.3390/ma16103722

**Published:** 2023-05-14

**Authors:** Guillermo Monrós, Mario Llusar, José A. Badenes

**Affiliations:** Department of Inorganic and Organic Chemistry, Jaume I University of Castellón, 12006 Castelló de la Plana, Spain; mllusar@uji.es (M.L.); jbadenes@uji.es (J.A.B.)

**Keywords:** clinobisvanite, ceramic pigment, high NIR reflectance, photocatalysis, sol–gel

## Abstract

Clinobisvanite (monoclinic scheelite BiVO_4_, *S.G.I2/b*) has garnered interest as a wide-band semiconductor with photocatalyst activity, as a high NIR reflectance material for camouflage and cool pigments and as a photoanode for PEC application from seawater. BiVO_4_ exists in four polymorphs: orthorhombic, zircon-tetragonal, monoclinic, and scheelite-tetragonal structures. In these crystal structures, V is coordinated by four O atoms in tetrahedral coordination and each Bi is coordinated to eight O atoms from eight different VO_4_ tetrahedral units. The synthesis and characterization of doped bismuth vanadate with Ca and Cr are studied using gel methods (coprecipitated and citrate metal–organic gels), which are compared with the ceramic route by means of the UV–vis–NIR spectroscopy of diffuse reflectance studies, band gap measurement, photocatalytic activity on Orange II and its relation with the chemical crystallography analyzed by the XRD, SEM-EDX and TEM-SAD techniques. The preparation of bismuth vanadate-based materials doped with calcium or chromium with various functionalities is addressed (a) as pigments for paints and for glazes in the chrome samples, with a color gradation from turquoise to black, depending on whether the synthesis is by the conventional ceramic route or by means of citrate gels, respectively; (b) with high NIR reflectance values that make them suitable as fresh pigments, to refresh the walls or roofs of buildings colored with them; and (c) with photocatalytic activity.

## 1. Introduction

Monoclinic BiVO_4_ (*I2/b*) is a direct band gap (2.4 eV) semiconductor with proper alignment with water oxidation potential that is applied to seawater splitting in the presence of a high ion concentration and in a corrosive environment [1]. BiVO_4_ exists in four polymorphs: orthorhombic (o-BiVO_4_) or pucherite, tetragonal-zircon type (t-BiVO_4_, *I41/amd*), monoclinic-scheelite or clinobisvanite (m-BiVO_4_, group *I2/b*) and scheelite-tetragonal. Although orthorhombic is the most common phase in nature (mineral pucherite), it has not been synthesized in the laboratory. The low-temperature synthesis of BiVO_4_ produces the zircon-tetragonal phase with a band gap of 2.9 eV, which, at 528 K, transforms into the monoclinic phase, reversible to tetragonal by adjusting the temperature. In these crystal structures, V is coordinated by four O atoms in a tetrahedral coordination and each Bi is coordinated to eight O atoms [2]. The BiO_8_ polyhedron known as bisdisphenoid or dodecadeltahedron is rather familiar in this case because all polymorphs of described BiVO_4_ show this typical coordination in zircon and scheelite; this polyhedron has 12 triangular faces, 18 edges and, of course, 8 vertices with symmetry D_2d_ in Schönflies notation [3].

Among non-titania (TiO_2_)-based visible-light-driven photocatalysts, monoclinic BiVO_4_ has proved to be an excellent material for photocatalytic water splitting and the photocatalytic degradation of organic compounds [4], as a high NIR reflectance material for camouflage and cool pigments and as a photoanode for PEC application from seawater [5].

BiVO_4_ is not classified as hazardous and is used to replace the highly toxic yellow cadmium sulfoselenide in ecofriendly paints applications [6]. In general, bismuth compounds do not show classified harmful properties and are used in medical application,, e.g., the so-called milk of bismuth [7] and or the bismuth subsalicylate (or hydrolyzed bismuth salicylate (Bi(C_6_H_4_(OH)CO_2_)_3_)), an antacid and anti-diarrheal agent [8].

BiVO_4_ can be stabilized in different crystal structures, yet high PEC efficiency is found only for the *n*-type-doped monoclinic phase. The theoretical description of the monoclinic polymorph is difficult because of spontaneous transformation to a tetragonal phase. The cause of instability of the monoclinic phase which spontaneously transforms to a tetragonal phase has not solved yet. Laraib et al. [9] suggest a crucial role of doping in the structure and, thus, the photoelectrochemical performance of BiVO_4_ because m-BiVO_4_ is related to the higher photoelectrochemical activity.

A simple review of the literature about the studied systems has been carried out analyzing the entries in WOS (Web of Science) and Google Academics. For the topic “calcium-doped BiVO_4_”, there are three entries (two in WOS and another one in Google): one for visible-light-activated (VLA) photocatalysis applied to the degradation of pharmaceutical contaminants of emerging concern and two for photoanodes for water splitting, all based in the semiconductor characteristics of Ca-BiVO_4_ solid solutions. Fatwa et al. [10] introduces for the first time calcium as an acceptor-type dopant into BiVO_4_ photoelectrodes, and the resulting Ca-doped BiVO_4_ photoelectrodes show anodic photocurrents with an enhanced carrier separation efficiency. BiVO_4_ and Ca-BiVO_4_ show indirect semiconductor behavior with band gap measured by a Tauc plot of 2.48 and 2.44 eV, respectively. Li et al. [11] prepare micropowders of M^II^_x_Bi_1−x_V_1−x_Mo_x_O_4_ (M^II^ = Ca, Sr, x = 0.1 to 0.9) by the ceramic method; the samples x = 0.1 in the cases of samples containing calcium and strontium afford the highest water-splitting performance.

In the case of “chromium-doped BiVO_4_”, twelve entries are detected—nine in WOS and two in Google. Seven are related to different photocatalytic systems applied to the photoreduction of chromium (VI) in wastewater and the other four to photoelectrochemical water splitting. The Cr-doped BiVO_4_ are synthesized using chromates via liquid–solid state reaction obtaining Cr(VI) solid solutions in BiVO_4_, where Cr(VI) replaces V(V). Krysiak et al. [12] studied photoanodes for water splitting prepared by spray printing of molybdenum-doped BiVO_4_ using a liquid–solid state reaction in ethanol media by means of multicomposite catalysts containing nickel, iron, and chromium introduced as nitrates. The formed catalyst particles were not uniformly incorporated into the Mo-BiVO_4_ structure, but rather creating agglomerates. This suggests phase segregation, which may be beneficial for photoelectrochemical activity. Okuno et al. [13] prepared Cr^6+^-BiVO_4_ solid solutions from Bi(NO_3_)_3_.5H_2_O, V_2_O_5_ and CaCrO_4_ in HNO_3_. The resulting solution was stirred for 26–168 h in the dark and the resulting precipitate was washed and dried at 80 °C for 5 h. The single phase of monoclinic scheelite BiVO_4_ was obtained with up to 2% Cr doping while mixtures of monoclinic scheelite and tetragonal-zircon BiVO_4_ were obtained for samples with larger amounts of doping (3–5%) even with a prolonged reaction time of up to 4 weeks.

Therefore, the literature related to Ca- and Cr-doped BiVO_4_ applies to the photoelectrochemical splitting of water and uses solid-state liquid reactions in organic media to preserve high valence ions such as Cr(VI). Therefore, it is not applicable for the synthesis of ceramic pigments that need high temperatures to be used in ceramic glazes and earthenware (1000–1300 °C) in construction applications.

This work addresses the synthesis and characterization of bismuth vanadate doped with calcium or chromium with various functionalities (a) as pigments for paints and enamels; (b) with high NIR reflectance values that make them suitable as cool pigments, to refresh the walls or floors of buildings colored with them; and (c) with photocatalytic activity.

## 2. Materials and Methods

### 2.1. Synthesis Methods

The synthesis methodology will be based on the ceramic method CE, the ammonia coprecipitation method CO and the metal–organic decomposition method MOD.

Ceramic samples (CE) were synthesized from Cr_2_O_3_, α-Bi_2_O_3_ and NH_4_VO_3_ 99.9 wt% supplied by Sigma-Aldrich. These precursors with a particle size between 0.3 and 5 µm were mechanically homogenized in an electric grinder (20,000 rpm) for 5 min and the mixture fired at the corresponding temperature and soaking time.

CO and MOD samples were synthesized from Cr(NO_3_)_3_.9H_2_O, NH_4_VO_3_ (previously dissolved in HNO_3_ 30 wt%) and Bi(NO_3_)_3_.5H_2_O (99.9 wt%, Sigma-Aldrich, St. Louis, MO, USA). For 5 g of the product, these precursors were dissolved in 200 mL of water, then citric acid is added with a molar ratio of Bi:Acid = 1:x (x = 0 or CO sample, 0.25, 1, 2). This solution was continuously stirred at 70 °C and ammonia 17.5 wt% was dropped until gelation occurred at approximatively pH 7.5. The gel was dried at 110 °C and fired at the corresponding temperature and soaking time (500 °C/1 h for charring CO and MOD gels, and 600 °C/3 h for CE and the previously charred CO and MOD gels).

### 2.2. Characterization Methods

X-ray diffraction (XRD) was carried out on a Siemens D5000 diffractometer (München, Germany) using Cu K_α_ radiation (10–70°2θ range, scan step 0.02°2θ, 4 s per step and 40 kV and 20 mA conditions).

L*a*b* and C*h* color parameters of glazed samples were measured following the CIE-L*a*b* (Commission Internationale de l’Éclairage) colorimetric method using a X-Rite SP60 spectrometer, with standard lighting D65 and a 10° observer [14]. L* measures the lightness (100 = white, 0 = black) and a* and b* the chroma (−a* = green, +a* = red, −b* = blue, +b* = yellow). C* (chroma) and h* (hue angle) can be estimated from a* and b* parameters by the Equations (1) and (2), respectively:C* = (a^2^ + b^2^)^1/2^(1)
h* = arctan (b*/a*)(2)

We can point out that the optimal chroma is obtained when lightness L* (measured by the L*a*b* method) is high for yellow (80–90), cyan (75–85) and green (70–80), middle for magenta (50–60) and red (45–55) and low for blue (25–35) hues.

UV–vis–NIR spectra of fired powder and the applications of the pigments were carried out by a Jasco V670 spectrometer (Madrid, Spain) through the diffuse reflectance technique, measuring absorbance (A in arbitrary units) or reflectance (R(%)) through the Kubelka–Munk transformation model. A band gap energy of semiconductors was calculated by a Tauc plot from the UV–vis–NIR diffuse reflectance spectra [15].

The solar reflectance, *R* or total solar reflectance, R_NIR_ or solar reflectance in the NIR and R_Vis_ solar reflectance in the visible spectrum are evaluated from UV–vis–NIR spectra, through the diffuse reflectance technique, as the integral of the measured spectral reflectance and the solar irradiance divided by the integral of the solar irradiance in the range of 350–2500 nm for *R*, 700–2500 nm for R_NIR_ and 350–700 nm for R_Vis_ as in the Equation (3):(3)$$R=\frac{\int_{350}^{2500}r\left(\lambda \right)i\left(\lambda \right)d\lambda }{\int_{350}^{2500}i\left(\lambda \right)d\lambda }$$
where (a) *r*(*λ*) is the spectral reflectance (Wm^−2^) measured from UV–vis–NIR spectra and (b) *i*(*λ*) is the standard solar irradiation (Wm^−2^ nm^−1^) according to the American Society for Testing and Materials (ASTM) Standard G173-03.

Microstructure characterization of powders was carried out by scanning electron microscopy (SEM) using a JEOL 7001F electron microscope (Akishima, Japan) and transmission electron microscopy (TEM) using a HITACHI electron microscope and selected area electron diffraction (SADP) (following conventional preparation and imaging techniques).

The photocatalytic tests were performed using a dispersion of 500 mg/L of powder added to a solution 0.6·10^−5^ M of Orange II in pH 7.42 phosphate buffer media (NaH_2_PO_4_·H_2_O 3.31 g and Na_2_HPO_4_·7H_2_O 33.77 g solved in l.000 mL of water). The degradation of Orange II in buffer media was followed by colorimetry at *λ* = 485. The UV irradiation source was a mercury lamp of 125 W emitting in the range 254–365 nm. The suspension was first stirred in the dark for 15 min to reach the equilibrium sorption of the dye. Aliquot samples were taken every 15 min to measure the change in the dye concentration. Control experiments with Orange II solution and without catalyst, were conducted before the photocatalytic experiments (CONTROL). Commercial anatase P25 from Degussa was used as a reference to compare its photocatalytic activity with the studied samples [16].

The photodegradation curves were analyzed following the Langmuir–Hinshelwood model [16] measuring parameters such as the degradation half-time t_1/2_ and the correlation of kinetic data R^2^.

In order to analyze the pigmenting capacity of powders as ceramic pigments, the powders were 3 wt% mixed with a double-firing frit (composition in oxides (wt%); SiO_2_(62.3), B_2_O_3_(9.1), Al_2_O_3_(10.3), CaO(5.9), K_2_O(9.3), ZnO(3.1), ZrO_2_(0.1)) supplied by Alfarben S.A., with a maturation point at 1050 °C that devitrifies zircon. The mixture of the frit and the pigment were fired applying a standard firing cycle used in the ceramic tile industry, maintaining the maximum of temperature (1050 °C) for 5 min.

## 3. Discussion

A cooperative transition (COT) mechanism is assumed [3]. In this mechanism, the chromophore enters the solid solution, replacing an isovalent cation but with dissimilar size; if the chromophore is bigger than the replaced ion, it becomes “compressed” even in small concentrations and even more if the incorporation of the chromophore ion M occurs, with increasing covalence and polarizability of the M-O bond. The M-O distance decreases and the high tetragonal distortion of d orbitals of the transition ion enhances the crystalline field over de ion. Consequently, absorption bands shift to higher frequencies and the color shifts to blue. If the chromophore is smaller than the replaced ion, it becomes “relaxed” in the lattice site (in this case, Shannon–Prewitt Crystal Radii [17]: Bi^3+^(VIII) = 1.31 Å, Ca^2+^(VIII) = 1.26 Å, Cr^3+^(VIII) = 0.7 Å) and the crystalline field over the d orbitals goes down; therefore, the absorption bands shift to higher wavelengths. In short, in this mechanism, there is a cooperative transition between the chromophore and the replaced ion that modifies the crystalline field over the chromophore ion.

In this case, the replacement of Bi^3+^ by the smaller Ca^2+^ or Cr^3+^ should relax the dopant ions, which shifts its absorption to higher wavelengths, but aliovalent replacement with Ca^2+^ probably increases the concentration of charge-carrying mobile defects of oxygen [13,18].

### 3.1. The Effect of the Dopant Concentration

Figure 1 shows the powder samples (Ca_x_Bi_1−x_)VO_4_, x = 0, 0.1, 0.2, 0.4 (600 °C/3 h): (a) aspect by binocular lens (×40), and its L*a*b*, and (b) 3 wt% washed samples glazed in a double-firing frit (1050 °C). The entrance of Ca^2+^ replacing Bi^3+^ is associated with a slight decrease in color intensity (lightness L* and yellow hue b* increase, but red hue a* decreases). The pigments are dissolved, unstabilized in the glaze, and do not produce coloration (Figure 1b).

Figure 2 shows the UV–vis–NIR spectra of calcium-doped samples (Ca_x_Bi_1−x_)VO_4_, x = 0.1, 0.2, 0.4 (600 °C/3 h), showing a transference charge band centered at 400 nm with a typical semiconductor performance. Table 1 shows the main characterization of (Ca_x_Bi_1−x_)VO_4_ (600 °C/3 h) samples: the band gap measured from UV–vis–NIR spectra by the Tauc procedure (Figure 3) indicates that the semiconductors show a direct performance and the band gap increases slowly from 2.25 eV for pure BiVO_4_ (x = 0) to 2.31 eV for the x = 0.4 doped sample (10). Thereby, NIR reflectance also increases continuously with x: from 53% for pure BiVO_4_ to 62% for the x = 0.4 sample, improving the cool performance of BiVO_4_.

All XRD of these Bi_1−x_Ca_x_VO_4_ ceramic samples show clinobisvanite (V) as a single phase, except the x = 0.4 sample, which shows residual peaks of CaV_2_O_6_ (see Figure 4). Therefore, it can be pointed out that the limit of the solid solution of Ca^2+^ in Bi_1−x_Ca_x_VO_4−x/2_ is lower than x = 0.4 and stabilizes the monoclinic polymorph (11).

Figure 5 shows the ceramic CE powders with chromium (Cr_x_Bi_1−x_)VO_4_, x = 0, 0.1, 0.2, 0.4 fired at 600 °C/3 h). Figure 5a shows the aspect through binocular lens (×40) and L*a*b* of yellow–green-colored samples; macroaggregates of homogeneous microcrystals can be observed and the entrance of chromium produces a more intense color (L* decreases with x) and less red and yellow hues (both a* and b* parameters decrease with x).

When washed powders were 3 wt% glazed in a double-firing frit (1050 °C) (Figure 5b) an interesting turquoise color is developed by chromium-doped samples, more intense with the chromium amount x: L* decreases, green (negative a*) decreases and blue (negative b*) increases, respectively, resulting x = 0.4 (L*a*b* = 53.8/−5.8/−8.9). The optimum turquoise sample. Figure 5 shows the XRD of chromium doped Bi_1−x_Cr_x_VO_4_ ceramic samples: all samples show clinobisvanite (V on figure) as majoritarian phase with residual peaks associated with hexagonal Cr_2_O_3_ (C) and Scherbinite V_2_O_5_ (S), which slightly increase in intensity with the amount of doping agent x (best observed in the highly doped sample (x = 0.4)).

The UV–vis–NIR spectra of (Cr_x_Bi_1−x_)VO_4,_ x = 0.1, 0.2, 0.4 (600 °C/3 h) powders (see spectra on Section 3.2 and Section 3.3 for sample x = 0.4) show optical absorption bands that can be associated with Cr^3+^ (3d^3^) in the dodecadeltahedral position (D_2d_ symmetry group), with a multiband at 220–520 nm integrated almost for the overlap of bands centered at 270, 380 and 480 nm, respectively, and a shoulder at 750 nm (that confers a semiconductor appearance to the spectrum) with bands at 680 and a shoulder at 760 nm. The electronic configuration for Cr^3+^(d^3^) may be described as d _x_^2^_−y_^21^d _z_^21^ d_xz,yz_^1^d_xz_d_xy_ [19]. Table 1 shows the characterization of these chromium-doped (Cr_x_Bi_1−x_)VO_4_ (600 °C/3 h) samples: the band gap measured from UV–vis–NIR spectra by the Tauc procedure indicates that the semiconductor shows a direct performance, decreasing its band gap slowly from 2.25 eV for pure BiVO_4_ (x = 0) to 2.13 eV for the x = 0.4 doped sample. Likewise, NIR reflectance also decreases continuously with x: from 53% for pure BiVO_4_ to 41% for the x = 0.4 sample, reducing the cool performance of BiVO_4_.

Figure 6 shows the UV–vis–NIR spectra of chromium-doped (Cr_x_Bi_1−x_)VO_4,_ x = 0.1, 0.2, 0.4 (600 °C/3 h) glazed samples. It is detected as an intense band centered at 270 nm to the charge transfer band of the frit used in the glaze, and relatively intense bands in the visible range at 380, 500 and 620 with a shoulder at 720 nm; a minimum of absorbance at 520 nm is detected, indeed a maximum of light reflected, that explains the turquoise color of the sample. These bands can be considered the same as that of the powders shifting to lower wavelengths.

Figure 7 shows the Tauc plots of chromium-doped (Cr_x_Bi_1−x_)VO_4,_ x = 0, 0.1, 0.2, 0.4 (600 °C/3 h) glazed samples showing a direct type semiconductor behavior. For the pure BiVO_4_ (x = 0) that unstabilizes in the glaze and produces white color associated with the glaze (see Figure 1b and Figure 2b), its Tauc plot indicates a direct band gap of 3.26 eV in the UV range due to the frit mixed with the pigment in the glaze. The band gap of turquoise glazes with chromium-doped BiVO_4_ increases slowly from 1.50 eV for x = 0.1 to 1.56 eV for the x = 0.4 sample in the vis–NIR range.

Since Ca-doped samples do not produce coloration and the solid solution is complete in this case, optimization studies with mineralizers and the use of the coprecipitation and MOD routes to obtain inks for glazes were performed only on chromium-doped samples.

### 3.2. The Effect of the Addition of Mineralizers in Cr-Doped Samples

Figure 8 shows the effect of the addition of three mineralizers in order to activate the synthesis improving the ionic diffusion on the system [20,21] of the optimum composition Cr_0.4_Bi_0.6_VO_4_: 3 wt% NH_4_Cl (min 1), 3 wt% 3NaF.2MgF_2_.Li_2_CO_3_ (min 2) and 2 wt% CaCO_3_.KCl +1 wt% NaF.Na_4_B_2_O_7_ (min 3). The DTA and TG analyses of the employed mineralizers (Figure 9) chosen with thermal activity at progressive temperatures show: (a) (min 1) melts and decomposes at 290 °C, showing an intense endothermic band (a very week endothermic band at 100 °C is detected also associated with water elimination) with an associated weight loss at TG of 100%; (b) (min 2) endothermic bands at 140, 460 and 625 °C as main DTA bands, and the band at 460 °C is associated with a weight loss of approximately 20% in TG pattern, due to carbonate decomposition, and finally (c) (min 3) shows the main endothermic bands at 140, 580 and 640 °C as main DTA bands with a weight loss in TG approximately 10% at 580 °C associated with carbonate decomposition.

Figure 8b shows the photographs of powders by binocular lens (×40) and the corresponding 3 wt% washed powders glazed in a double-firing frit (1050 °C) (L*a*b*parameters are included); the low-temperature mineralizer NH_4_Cl (min 1) is greenish (L* increases, a* decreases and b* increases) but the turquoise shade obtained in glazed samples is similar to the unmineralized sample, the middle mineralizer 2 produces green shades in both powder and glazed samples, and the higher-temperature mineralizer (min 3) is also greenish in powder but shows a high turquoise intensity in glazes (L* decreases and negative b* increases). The addition of mineralizers only shows a significant improvement of the turquoise shade of the glazed sample with mineralizer 3 with a near thermal activity (580 °C) to the firing employed temperature (600 °C) (20).

Figure 10 shows the XRD diffractogram of some mineralized samples: mineralizer 1 shows m-BiVO_4_ and residual peaks of S (scherbinite V_2_O_5_) and C (hexagonal Cr_2_O_3_). In the case of mineralizer 2, C (hexagonal Cr_2_O_3_) and very weak peaks of W (CrVO_4_) are detected; finally, for mineralizer 3 only residual peaks of W (CrVO_4_) are detected. Therefore, the relative high-temperature mineralizer 3 shows the cleaner diffractogram (with minimal residual phases) and the best turquoise result in the glaze.

Figure 11 shows the UV–vis–NIR of Cr_0.4_Bi_0.6_VO_4_ powders and glazed mineralized samples, respectively. As described above, powders show the optical absorption bands associated with Cr^3+^ (3d^3^) in the dodecadeltahedral position but the absorbance is higher in mineralized samples; and in the case of min 3, the shoulder at 680 nm practically disappears. Likewise, glazed samples show an intense band centered at 300 nm due to the charge transfer band of the frit used in the glaze and relatively intense bands in the visible range at 500, 610 and the shoulder at 680 nm; a minimum of absorbance at 500 nm is detected; indeed a maximum of light reflected, which explains the turquoise color of the sample. The band gap of powders (measured by the Tauc method) decreases for mineralized samples; 2.11 eV for min 1 and 2.09 eV for min 2 and min 3 (Table 2). On the other hand, the band gap associated with the shoulder at 680 nm for glazed samples indicates similar values for mineralizer 2 (1.56 eV) whereas those for mineralizers 1 and 3 are slightly lower (1.55 and 1.54 eV, respectively) (Table 2).

Finally, the reflectance R_Vis_ and R_NIR_ of powders decrease in all mineralized samples associated with a greenish turn and as the intensity of the color increases (lower L*) (Table 2). On the other hand, for glazed samples, the total reflectance is very similar in all mineralized samples and the CE sample (*R* = 43–44) (Table 2), in agreement with the very similar band gap and absorbance spectra on Figure 11.

### 3.3. The Effect of Non-Conventional Methods and Ink Application in Cr-Doped Samples

Figure 12 shows the coprecipitated sample (CO) and the effect of citric acid (MOD): (I,II) show the photographs of powders by binocular lens (×40) charred at 500 °C/1 h and successively fired at 600 °C/3 h, respectively, (III,IV) show the screen printing (48 threads/cm) of direct colloidal emulsion and the 500 °C/1 h powder mixed with diethylenglycol (weight ratio 2:5), and (V,VI) show the 3 wt% washed samples glazed in a double-firing frit (1050 °C) of powders fired at 500 °C/1 h and 600 °C/3 h, respectively (L*a*b*parameters are included).

The charred powders at 500 °C/1 h (Figure 12I) are of dark green shades in MOD samples associated with the remaining graphitic residue in the material. The stabilized samples fired at 600 °C/3 h (Figure 12II), free of carbon, are of yellow–green shades with L*a*b* of approximately 47/10/35, similar in all samples. The screen-printing depositions on ceramic support (Figure 12III,IV) show brown shades for CO and citric 0.25 samples and greenish shades for samples citric 1, 1.5 and 2 (b* approximately −2.5). Finally, glazed samples (Figure 12V,VI) show grey shades which darken with x: the sample with x = 2 shows a dark-grey (black) shade with very low chroma C* (L*a*b* = 41.6/−0.4/0.6 for 500 °C/1 h and 44.7/1.1/3.9 for stabilized sample 600 °C/3 h).

Figure 13 shows the XRD diffractograms of CE, CO (x = 0) and citric MOD samples fired at 500 °C/1 h and 600 °C/3 h, respectively. At 500 °C, only broad peaks of m-BiVO_4_ are detected. At 600 °C, as in the CE sample, the CO and MOD samples show clinobisvanite m-BiVO_4_ with residual peaks associated with S (scherbinite V_2_O_5_) and C (hexagonal Cr_2_O_3_); in the CO and MOD samples, the intensity of residual phases slightly decreases.

CE and mineralized yellow powders produce turquoise colors in the glaze and greenish CO and MOD powders give grey-black colors in the glaze. Figure 14 shows the UV–vis–NIR representative spectra of a yellow powder (CE sample, turquoise in glaze) and a greenish powder (citric x = 2 MOD sample, grey-black in glaze). As described above, the optical absorption bands associated with Cr^3+^ (3d^3^) in the dodecadeltahedral position (D_2d_ symmetry group) are detected in the case of the yellow powder (CE sample, turquoise in glaze). For the greenish powder (citric x = 2 MOD sample, grey-black in glaze), the spectrum is similar, but the absorbance in the 580–1500 nm range increases. Figure 15 shows the Tauc plots of these representative samples; the band gap increases from 2.13 eV for yellow powder to 2.20 eV for greenish MOD powder (Table 3).

Figure 16 shows UV–vis–NIR spectra of glazed samples of Cr_0.4_Bi_0_._6_VO_4_ composition: (a) CITRIC MOD glazed samples 500 °C/1 h; (b) CE and CITRIC x = 2 samples 600 °C/3 h. (including sample CE). Bands at 270, 320,420, 580 and 670 nm can be detected, showing a shift at lower wavelengths relative to ceramic samples and a significant increase in absorption to approximately 500 nm associated with the darkening of glazes. The direct band gap of the semiconductors measured from Tauc plots, as shown in Figure 17, for the edge of absorbance in the NIR range (3 in Figure 16b) shows that all MOD samples show a band gap of 1.53 eV slightly lower than the homologous CE sample (1.56 eV) (Table 3).

Finally, the R_Vis_ and R_NIR_ reflectance of the powders decrease slightly in CO and MOD samples associated with greenish hue (lower L*) (Table 3). On the other hand, for glazed samples, the reflectance is very similar in all CO and MOD samples, with a lower R_Vis_ than the CE sample associated with the darkening of samples, but higher R_NIR_ (Table 3), in agreement with the absorbance spectra on Figure 16 and despite the darkening of samples.

Figure 18 presents the photodegradation test over Orange II of powders (Cr_0.4_Bi_0_._6_VO_4_) fired at 600 °C/3 h (CE and CITRATE (x = 2) samples) with the Langmuir–Hinshelwood kinetics parameters t_1/2_ (degradation half-time and data correlation R^2^) which for the commercial reference P25 of Degussa and for the control (without powder addition) are t_1/2_/R^2^= 25 min/0.898 and 497 min/0.945, respectively. The ceramic samples show an interesting degradation half-time of 85 min that increases to 150 min for the citrate sample (lower than 497 min for the simple photolysis of the CONTROL test).

Figure 19 shows the SEM micrographs of representative powders of Cr_0.4_Bi_0_._6_VO_4_ composition: the CE powder, which produces a turquoise color on glazes, shows well-developed prismatic crystals of approximately 3 µm in length together with other irregular particles of 0.5 µm in size. The coprecipitated sample CO, which produces a grey color on glazes, shows very fine nanoparticles, forming agglomerates of 3–15 µm of size, finally MOD x = 0.25, which produces a grey color on glazes, shows microcrystals between 0.5 and 1 µm, forming big agglomerates of 10–20 µm. The decrease in the active surface due to the agglomeration of the fine particles of BiVO_4_ of the CO and MOD methods, observed by SEM, is probably associated with the decrease in the degradation half-time detected in the photodegradation test of these samples (12).

Figure 20 shows the TEM micrographs and selected area electron diffraction (SADP) of powders MOD x = 0.25 charred at 500 °C/1 h used in the preparation of inks IV; nanoparticles of 10–80 nm and a SADP compatible with polycrystals of clinobisvanite m-BiVO_4_ are detected.

## 4. Conclusions

Doped bismuth vanadate was synthesized using gel methods (coprecipitated and citrate metal–organic gels) and compared with the ceramic route.

The limit of the solid solution of Ca^2+^ in Bi_1−x_Ca_x_VO_4−x/2_ is lower than x = 0.4, in the case of Cr^3+^ fired at 600 °C, and residual Cr_2_O_3_ and V_2_O_5_ always remained, indicating an incomplete solid solution of chromium in clinobisvanite (m-BiVO_4_), but both cations stabilize the monoclinic polymorph. NIR reflectance increases continuously with Ca doping from 53% for pure BiVO_4_ to 62% (Eg = 2.31 eV) for Ca_0.4_Bi_0_._6_VO_4_ samples, improving the cool performance of BiVO_4_; conversely, the entrance of chromium decreases the NIR reflectance of clinobisvanite. The band gap of clinobisvanite increases for Ca-doped samples and decreases for Cr-doped samples in agreement with NIR reflectance evolution.

The yellow samples doped with Ca do not produce coloration in glazes but the yellow–green Cr-doped samples produce interesting turquoise shades in a double-firing frit (1050 °C) resulting in the optimum turquoise of the Cr_0.4_Bi_0_._6_VO_4_ sample (L*a*b* = 53.8/−5.8/−8.9) that maintains NIR reflectance at approximately 67% (band gap 2.13 eV). The use of mineralizer 3 (2 wt% CaCO_3_.KCl +1 wt% NaF.Na_4_B_2_O_7_) with a near thermal activity (580 °C) to the employed firing temperature (600 °C) produces a significant improvement in the turquoise shade of the glazed sample (L*a*b* = 49.6/−9.4/−9.4) with similar NIR reflectance (66%) and band gap (2.09 eV).

Cr-doped samples prepared by gel methods (CO and citrate MOD routes) produce yellow–green powders that darken with the amount of chromium, decreasing NIR reflectance and increasing the band gap. The corresponding glazed samples produce black shades with high NIR reflectance (69–71%) associated with a shift in the absorbance bands of Cr probably in dodecadeltahedral site to lower wavelengths relative to the ceramic sample and a significant increase in absorption at approximately 500 nm associated with the darkening of glazes.

The Cr-doped ceramic samples show an interesting degradation half time of 85 min that increases to 150 min for the citrate sample (lower than the 497 min for the simple photolysis of the CONTROL test).

To the best of our knowledge, Ca- or Cr-doped polyfunctional BiVO_4_ pigments have been prepared for the first time for paints, and for glazes in the case of chromium, producing colors from turquoise to black depending on whether the synthesis is a conventional ceramic route or through citrate gels, showing a high NIR reflectance that makes them suitable as cool pigments, to refresh the walls and roofs of buildings colored with them, and gives them photocatalytic activity.

## Figures and Tables

**Figure 1 materials-16-03722-f001:**
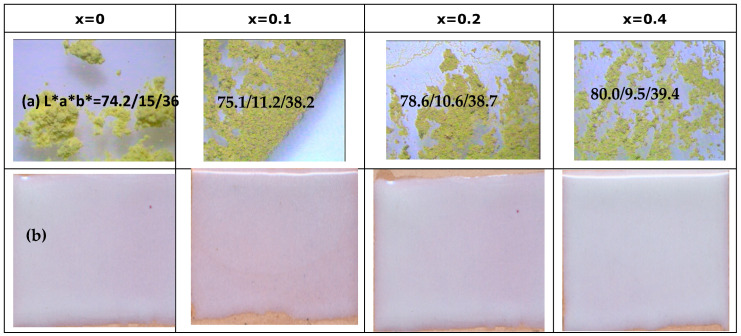
Powder samples (Ca_x_Bi_1−x_)VO_4,_ x = 0, 0.1, 0.2, 0.4 (600 °C/3 h): (**a**) aspect by binocular lens (×40), and L*a*b* of powders; (**b**) 3 wt% washed samples glazed in a double-firing frit (1050 °C).

**Figure 2 materials-16-03722-f002:**
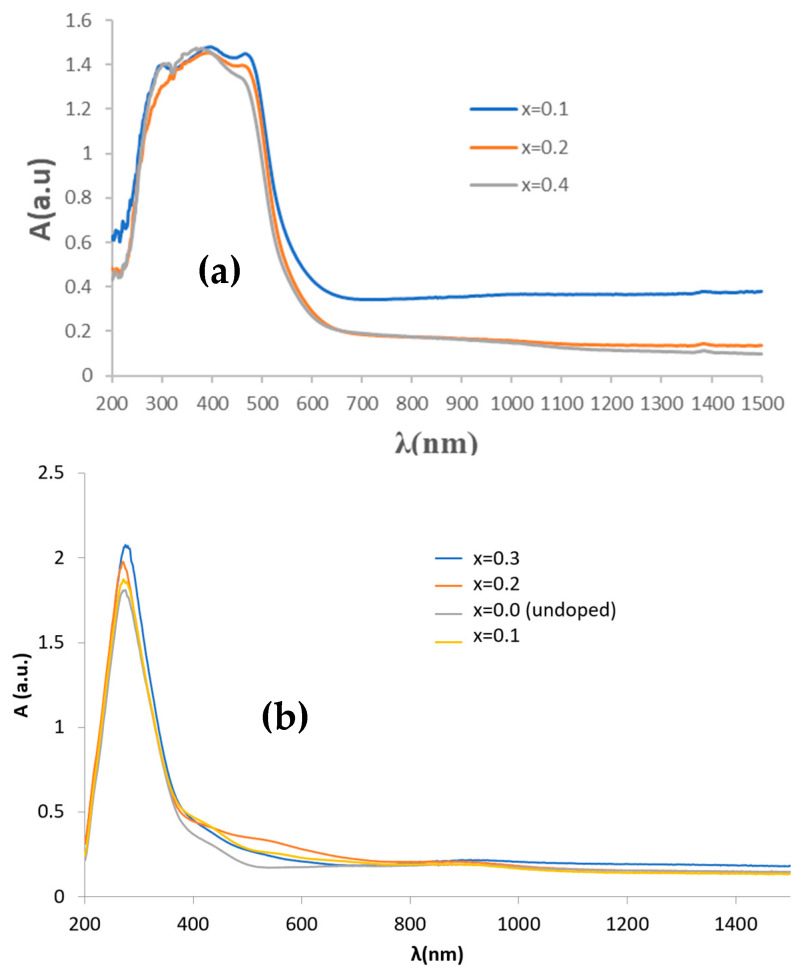
UV–vis–NIR spectra of (Ca_x_Bi_1−x_)VO_4,_ x = 0.1, 0.2, 0.4: (**a**) (600 °C/3 h) powders; (**b**) 3 wt% glazed samples.

**Figure 3 materials-16-03722-f003:**
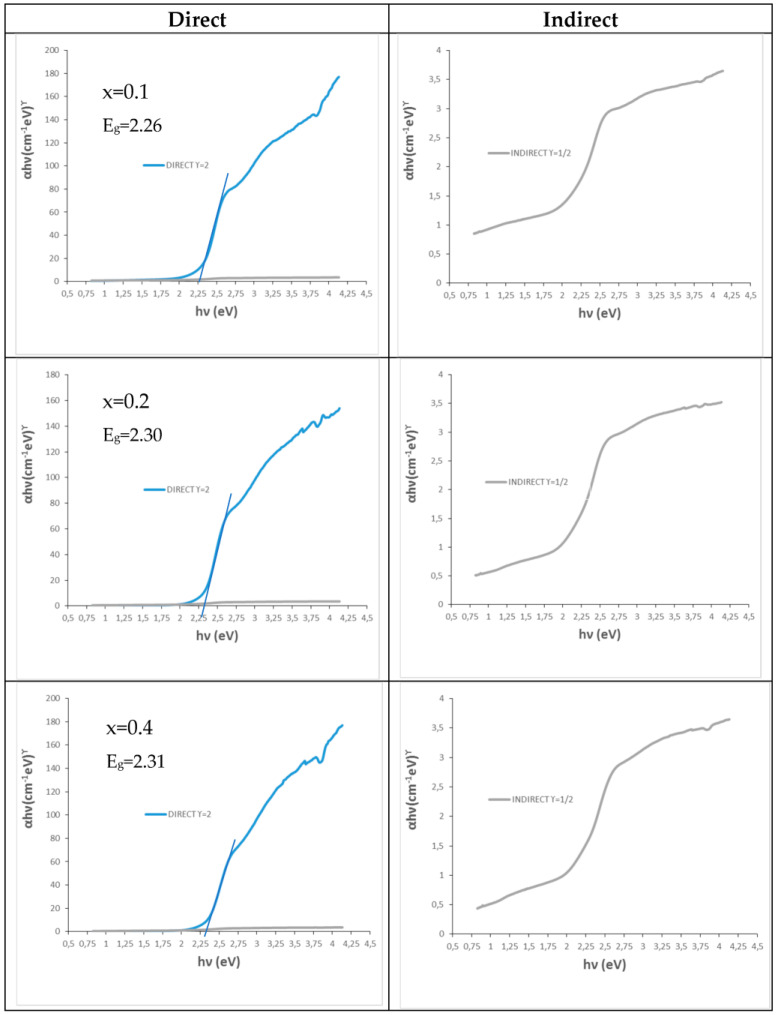
Tauc plots of (Ca_x_Bi_1−x_)VO_4_, x = 0.1, 0.2, 0.4 (600 °C/3 h) powders.

**Figure 4 materials-16-03722-f004:**
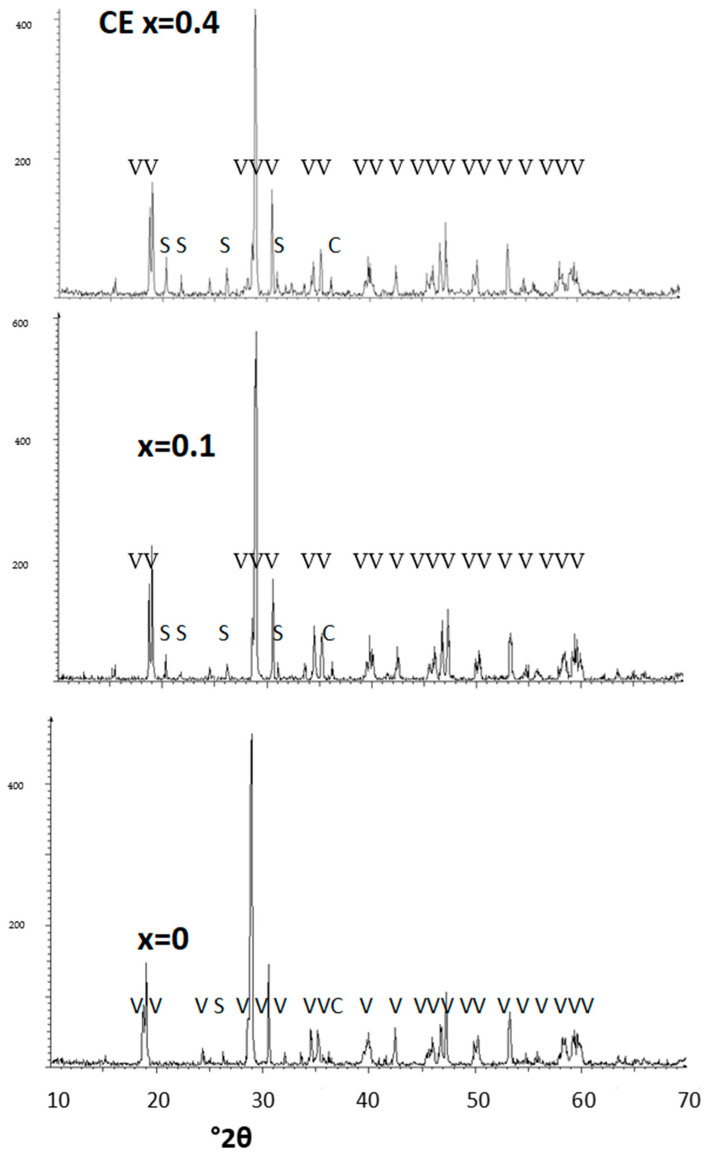
XRD of Bi_1−x_Cr_x_VO_4_ ceramic samples. Crystalline phases: V (m-BiVO_4_), C (hexagonal Cr_2_O_3_), and S (Scherbinite V_2_O_5_). The homologous Bi_1−x_Ca_x_VO_4_ ceramic samples show clinobisvanite (V) as a single phase except the x = 0.4 sample, which shows residual peaks of CaV_2_O_6_.

**Figure 5 materials-16-03722-f005:**
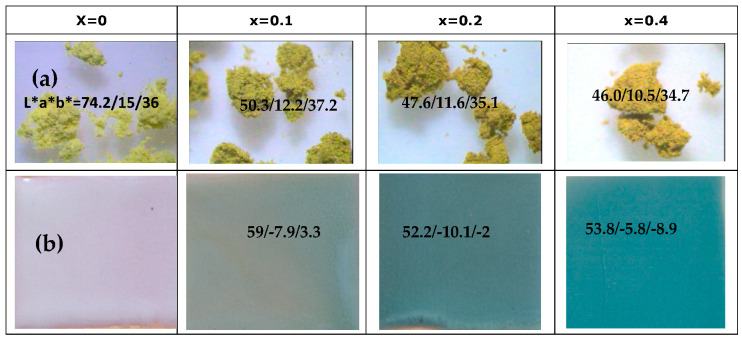
Samples (Cr_x_Bi_1−x_)VO_4,_ x = 0, 0.1, 0.2, 0.4 (600 °C/3 h): (**a**) aspect through binocular lens (×40), and L*a*b* of powders; (**b**) 3 wt% washed samples glazed in a double-firing frit (1050 °C).

**Figure 6 materials-16-03722-f006:**
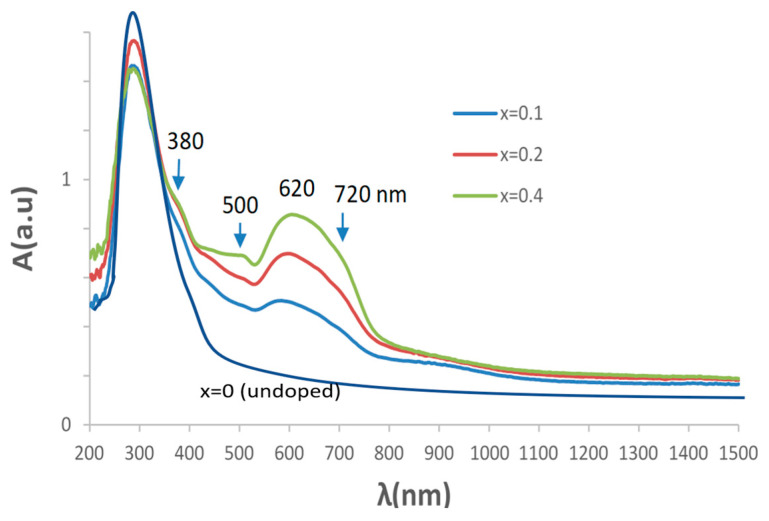
UV–vis–NIR spectra of Cr_x_Bi_1−x_VO_4_ glazed samples.

**Figure 7 materials-16-03722-f007:**
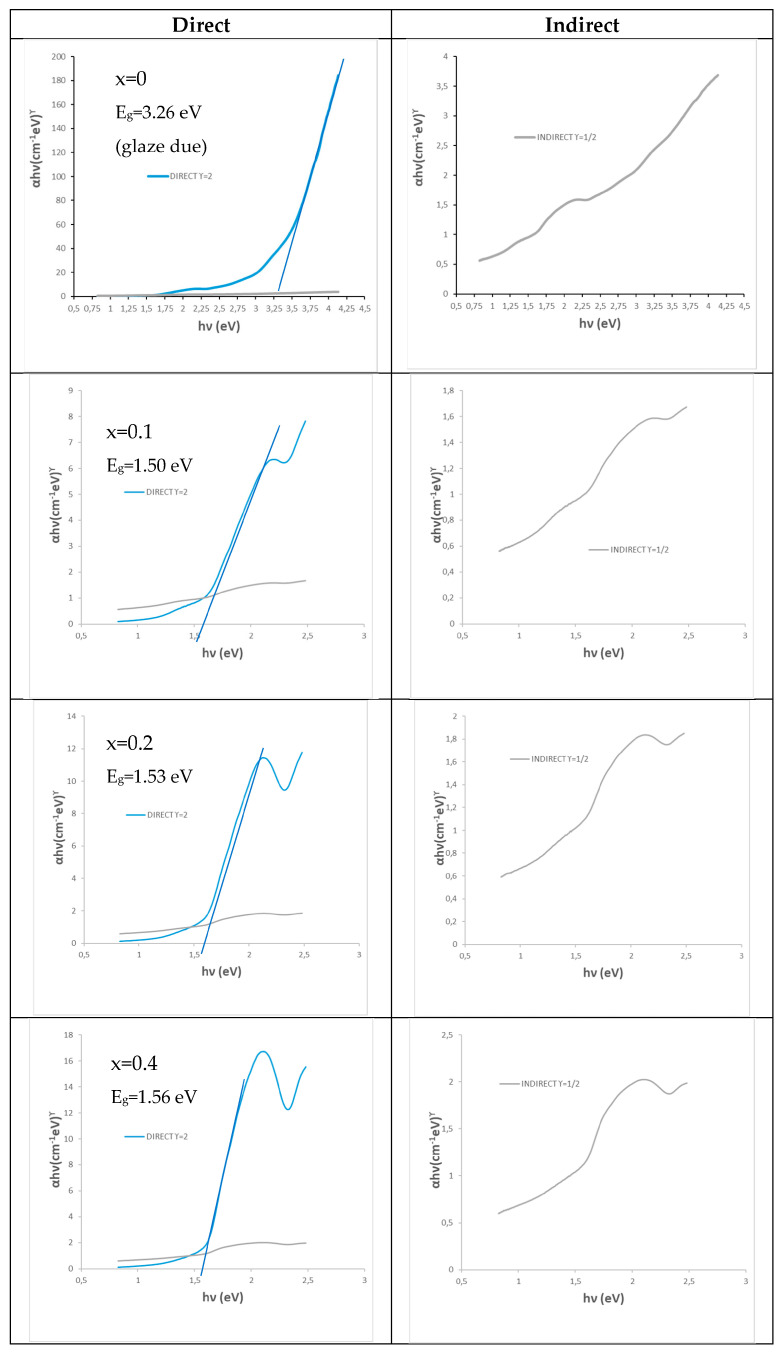
Tauc plots of (Cr_x_Bi_1−x_)VO_4,_ x = 0, 0.1, 0.2, 0.4 (600 °C/3 h) glazed samples.

**Figure 8 materials-16-03722-f008:**
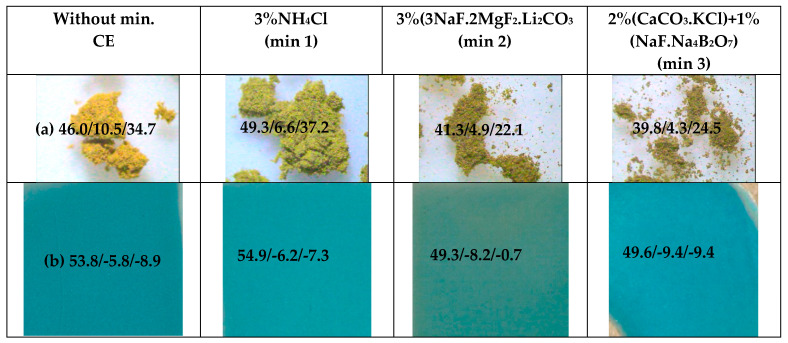
Effect of mineralizers on the Cr_0.4_Bi_0.6_VO_4_ sample: (**a**) photos of powders by binocular lens (×40); (**b**) 3 wt% washed samples glazed in a double-firing frit (1050 °C) (L*a*b*parameters are included).

**Figure 9 materials-16-03722-f009:**
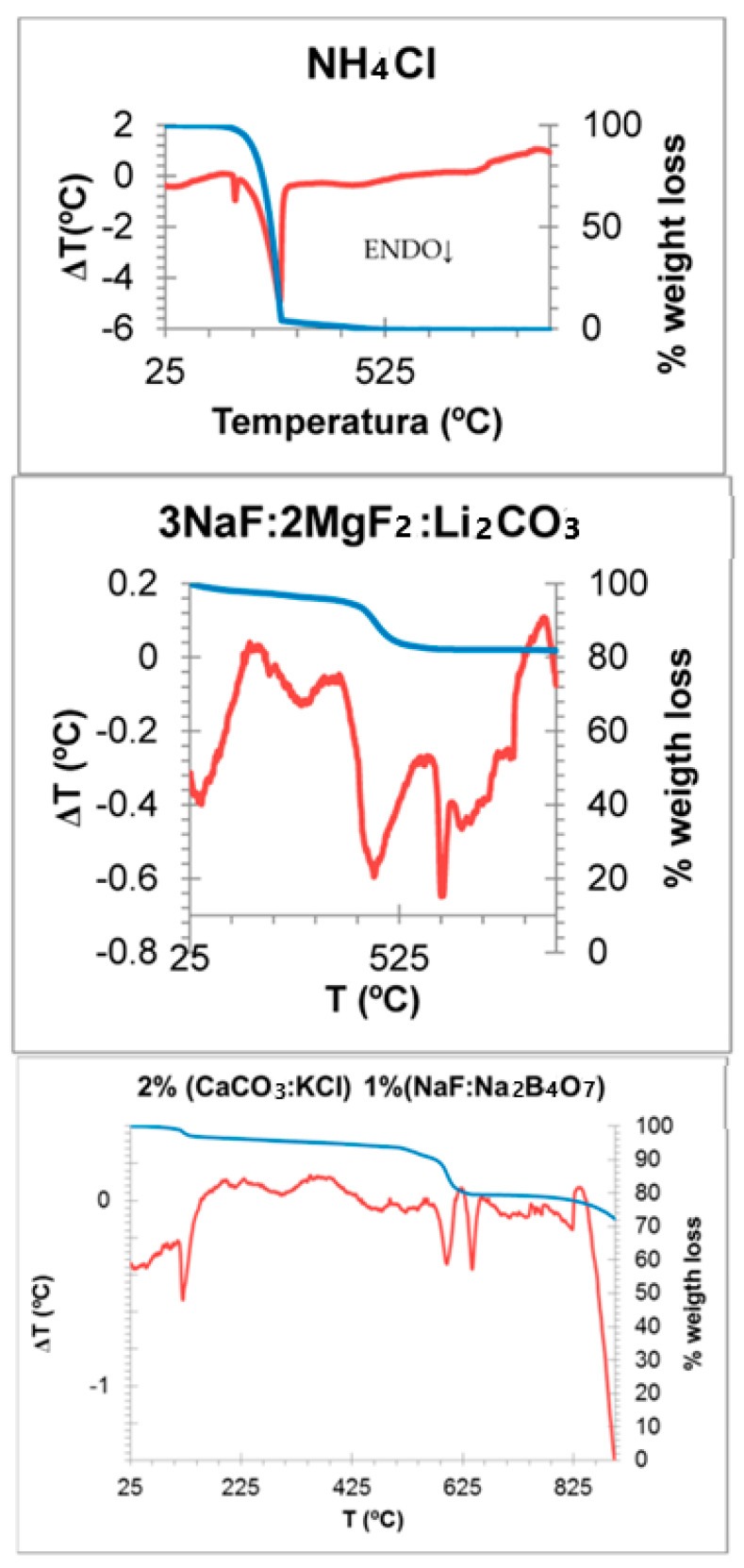
DTA (red) and TG (blue) of employed mineralizers.

**Figure 10 materials-16-03722-f010:**
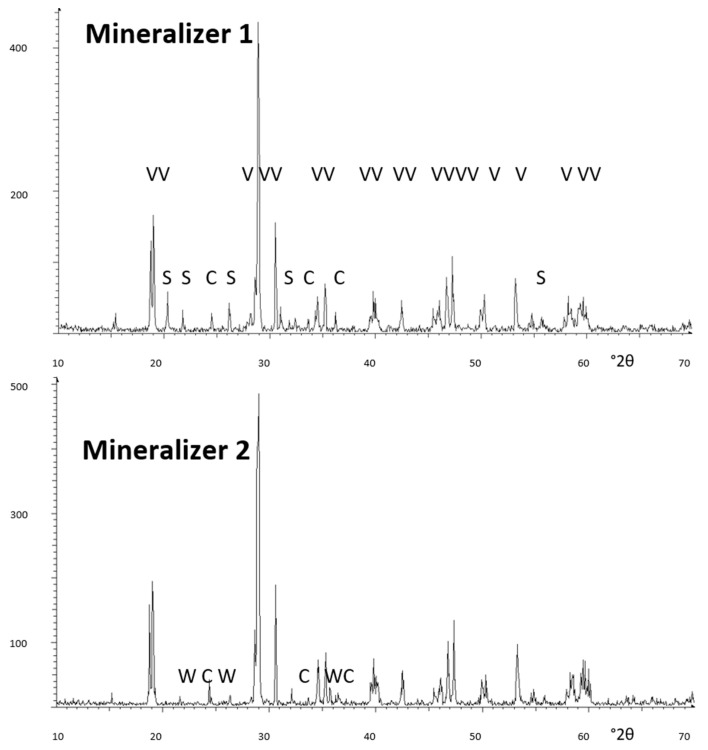
XRD of the Cr_0.4_Bi_0.6_VO_4_ mineralized samples. Crystalline phases: V (m-BiVO_4_), S (Scherbinite V_2_O_5_), C (hexagonal Cr_2_O_3_), and W (CrVO_4_).

**Figure 11 materials-16-03722-f011:**
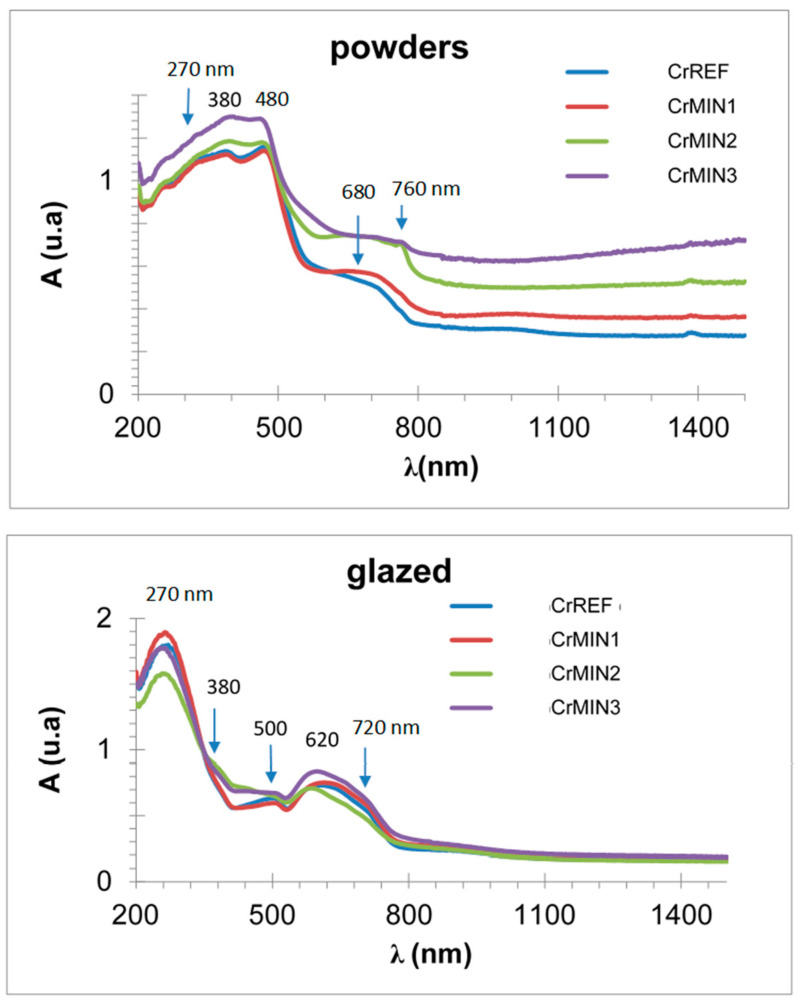
UV–vis–NIR spectra of the Cr_0.4_Bi_0.6_VO_4_ mineralized samples.

**Figure 12 materials-16-03722-f012:**
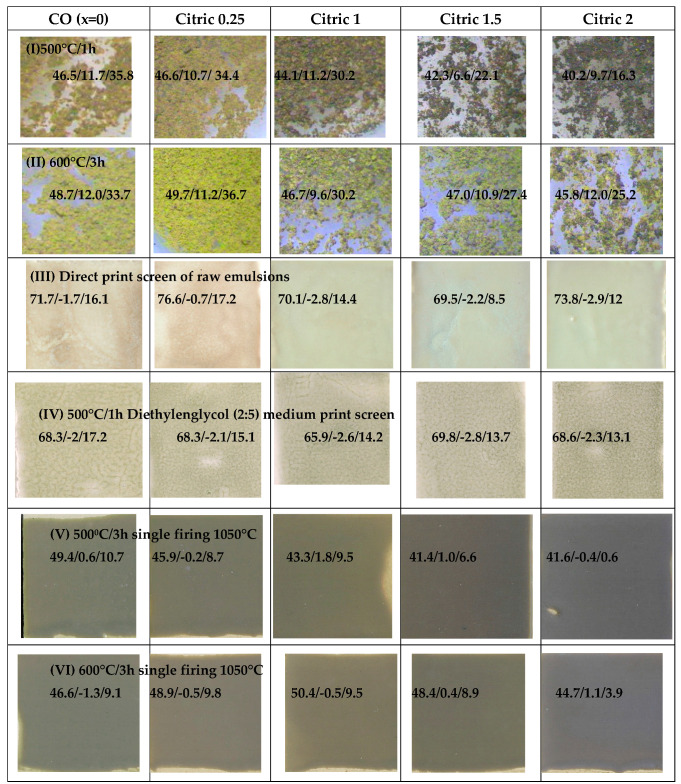
Coprecipitated sample (CO) and effect of citric acid (MOD) of Cr_0.4_Bi_0.6_VO_4_ composition: (**I**,**II**) photos of powders by binocular lens (×40) fired at 500 °C/1 h and 600 °C/3 h, respectively, (**III**,**IV**) print screen (48 threads/cm) of direct colloidal emulsion and the 500 °C/1 h powder mixed with diethylenglycol (weight ratio 2:5), and (**V**,**VI**) 3 wt% washed samples glazed in a double-firing frit (1050 °C) fired at 500 °C/1 h and 600 °C/3 h, respectively (L*a*b*parameters are included).

**Figure 13 materials-16-03722-f013:**
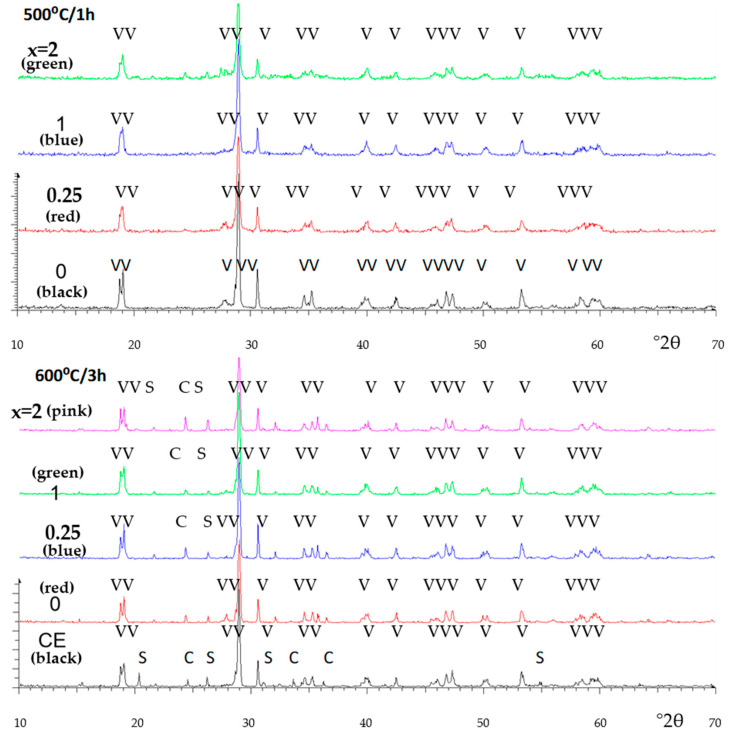
XRD spectra of CITRIC MOD samples of Cr_0.4_Bi_0.6_VO_4_ composition. Crystalline phases: V (m-BiVO_4_), S (Scherbinite V_2_O_5_), C (hexagonal Cr_2_O_3_).

**Figure 14 materials-16-03722-f014:**
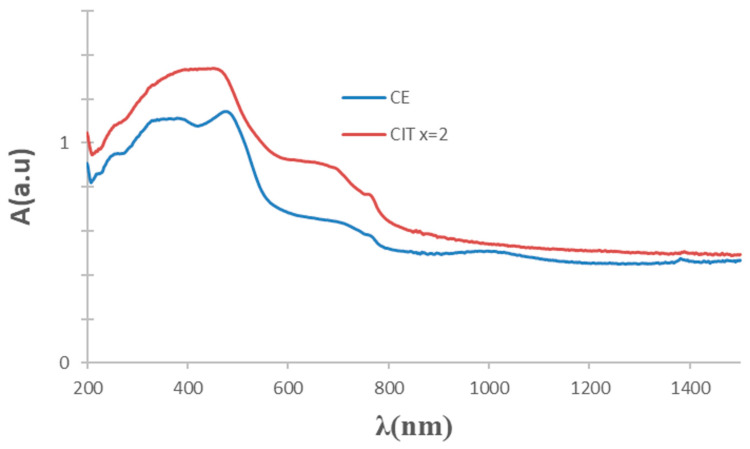
UV–vis–NIR spectra of CE and CITRIC x = 2 powder samples of Cr_0.4_Bi_0_._6_VO_4_ composition.

**Figure 15 materials-16-03722-f015:**
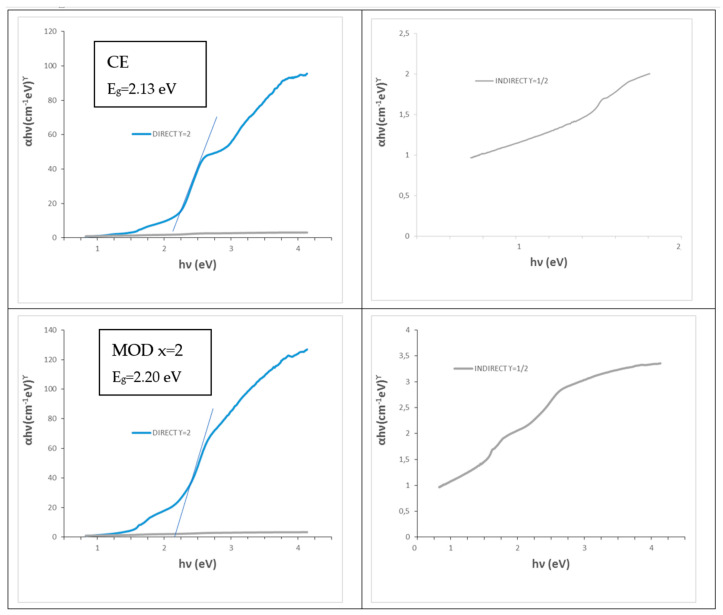
Tauc plots of CE and CITRIC x = 2 powders of Cr_0.4_Bi_0_._6_VO_4_ composition.

**Figure 16 materials-16-03722-f016:**
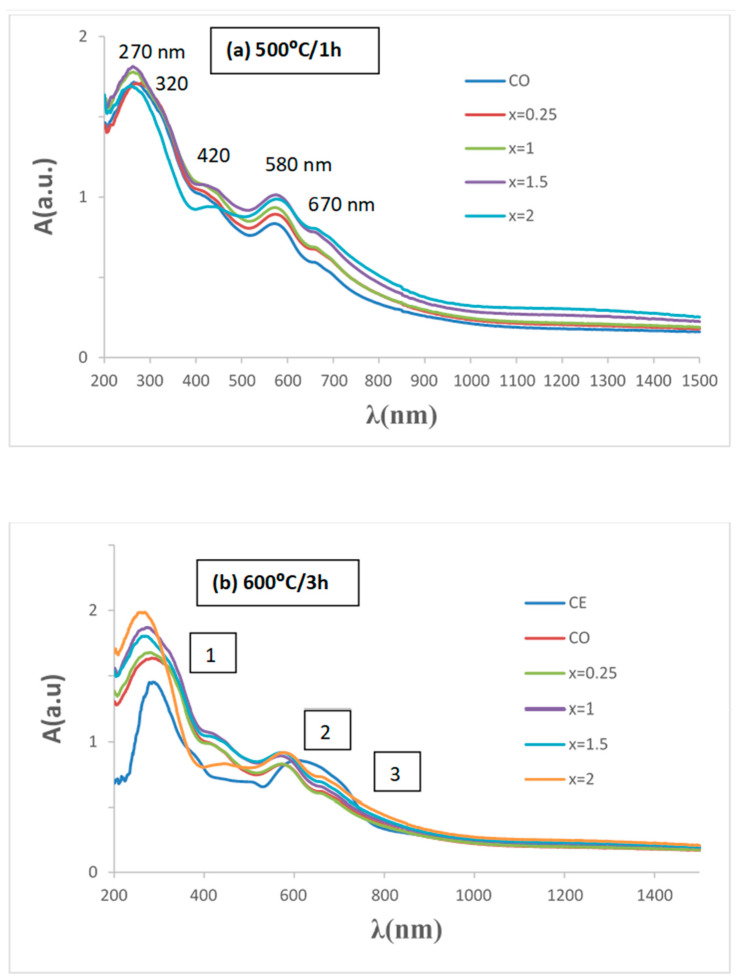
UV–vis–NIR spectra of glazed samples of Cr_0.4_Bi_0_._6_VO_4_ composition: (**a**) CITRIC MOD glazed samples 500 °C/1 h; (**b**) CE and CITRIC x = 2 samples 600 °C/3 h. (the analyzed absorbance inflection points are shown in the numbered boxes).

**Figure 17 materials-16-03722-f017:**
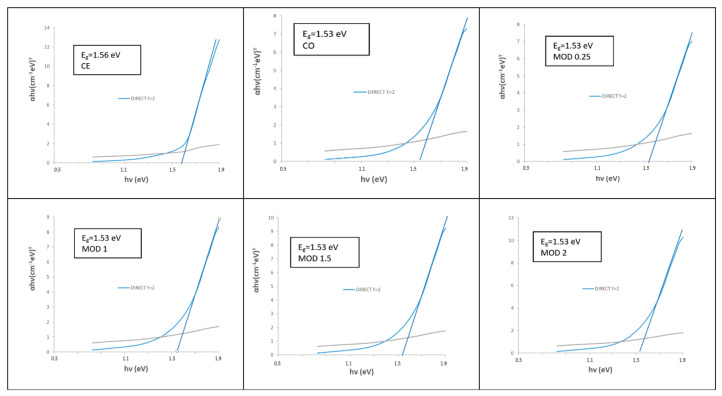
Tauc plots of glazed samples of Cr_0.4_Bi_0_._6_VO_4_ composition considering the absorption 3 on Figure 16b.

**Figure 18 materials-16-03722-f018:**
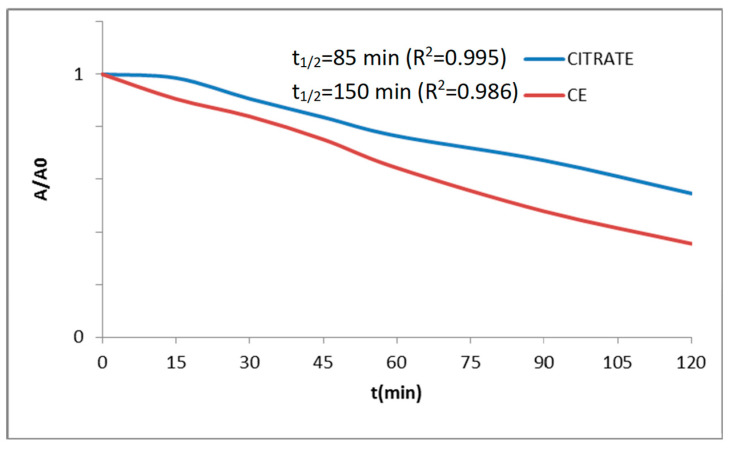
Photodegradation test over Orange II of powders (Cr_0.4_Bi_0.6_VO_4_) fired at 600 °C/3 h (CE and CITRATE (x = 2) samples) with the Langmuir–Hinshelwood kinetics parameters (degradation half-time t_1/2_ and data correlation R^2^).

**Figure 19 materials-16-03722-f019:**
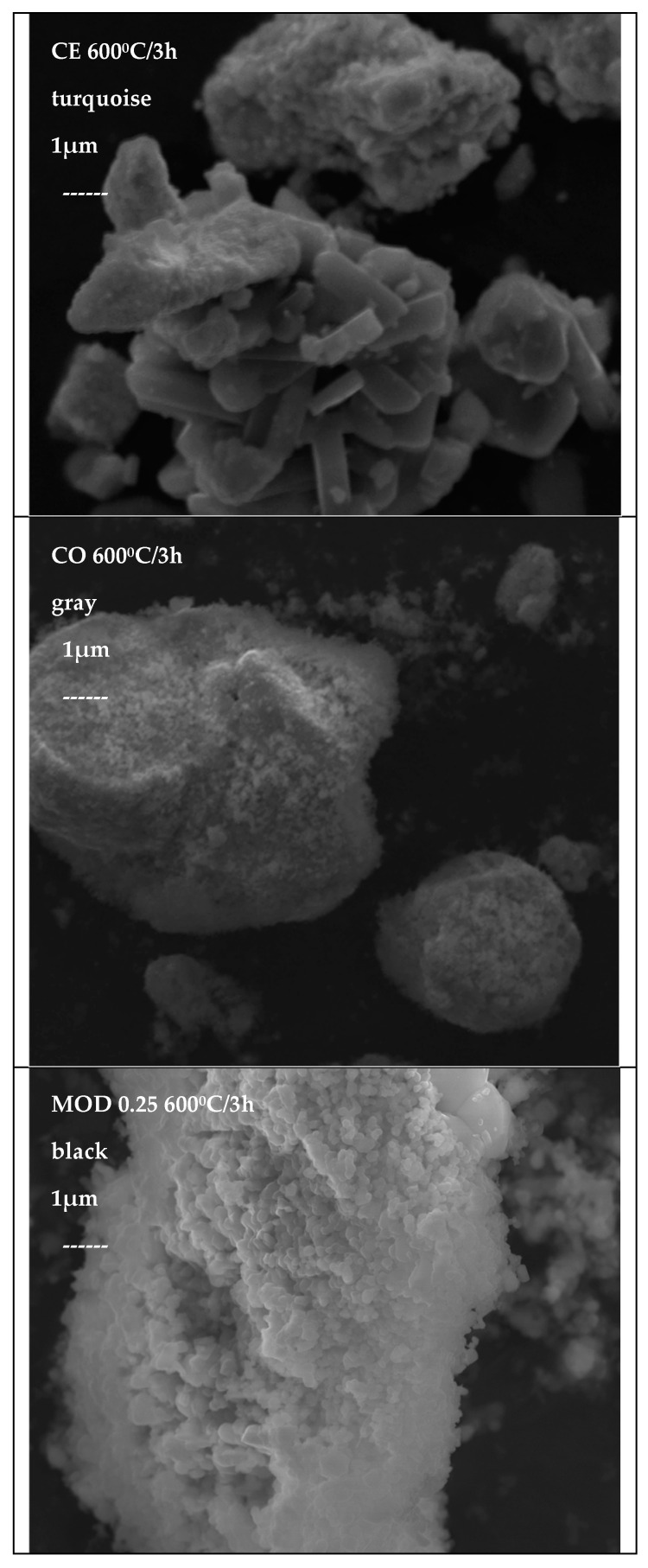
SEM micrographs of representative samples of Cr_0.4_Bi_0_._6_VO_4_ composition.

**Figure 20 materials-16-03722-f020:**
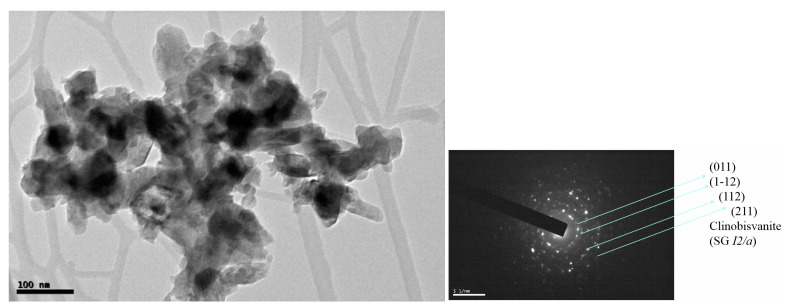
TEM micrographs and selected area electron diffraction (SADP) of powders MOD x = 0.25 (Cr_0_._25_Bi_0_._75_VO_4_) charred at 500 °C/1 h used in the preparation of inks IV.

**Table 1 materials-16-03722-t001:** Characterization of (M_x_Bi_1−x_)VO_4_, M = Ca, Cr_,_ x = 0, 0.1, 0.2, 0.4 (600 °C/3 h) samples.

	x = 0	x = 0.1	x = 0.2	x = 0.4
**(Ca_x_Bi_1−x_)VO_4_** **powder**				
L*a*b*	74.2/15.0/36.0	75.1/11.2/38.2	78.6/10.6/38.7	80.0/9.5/39.4
Eg (eV) (Tauc)	2.25	2.26	2.30	2.31
R_Vis_/R_NIR_/*R*	30/53/41	31/55/43	32/60/46	33/62/47
**(Cr_x_Bi_1−x_)VO_4_** **powder**				
L*a*b*	74.2/15/36	50.3/12.2/37.2	47.6/11.6/35.1	46.0/10.5/34.7
Eg (eV) (Tauc)	2.25	2.18	2.13	2.13
R_Vis_/R_NIR_/*R*	30/53/41	27/47/37	23/44/32	19/41/30
**(Cr_x_Bi_1−x_)VO_4_** **glazed**				
L*a*b*	white	59/−7.9/3.3	52.2/−10.1/−2	53.8/−5.8/−8.9
Eg (eV) (Tauc)	3.26	1.50	1.53	1.56
R_Vis_/R_NIR_/*R*	77/89/82	33/76/54	26/70/47	22/67/44

**Table 2 materials-16-03722-t002:** Characterization of mineralized samples of Cr_0.4_Bi_0.6_VO_4_ composition (600 °C/3 h).

	CE	min 1	min 2	min 3
**POWDERS**				
L*a*b*	46/10.5/34.7	49.3/6.6/37.2	41.3/4.9/22.1	39.8/4.3/24.5
E_g_ (eV) (Tauc)	2.13	2.11	2.11	2.09
R_Vis_/R_NIR_/*R*	19/41/30	18/39/28	16/37/26	13/35/21
**GLAZED**				
L*a*b*	53.8/−5.8/−8.9	54.9/−6.2/−7.3	49.3/−8.2/−0.7	49.6/−9.4/−9.4
E_g_ (eV) (Tauc)	1.56	1.55	1.56	1.56
R_Vis_/R_NIR_/*R*	22/67/44	22/68/44	21/67/44	23/66/43

**Table 3 materials-16-03722-t003:** Characterization of CO and MOD samples of Cr_0.4_Bi_0.6_VO_4_ composition (600 °C/3 h).

	CE	CO	MOD 0.25	MOD 1	MOD 1.5	MOD 2
**POWDERS**						
L*a*b*	46/10.5/34.7	46.5/11.7/35.8	46.6/10.7/34.4	44.1/11.2/30.2	42.3/6.6/22.1	40.2/9.7/16.3
E_g_ (eV) (Tauc)	2.13	2.14	2.14	2.16	2.18	2.20
R_Vis_/R_NIR_/*R*	19/41/30	18/38/28	17/39/28	16/37/27	15/38/27	14/38/26
**GLAZED (600 °C)**						
L*a*b*	53.8/−5.8/−8.9	46.6/−1.3/9.1	48.9/−0.5/9.8	50.4/−0.5/9.5	48.4/0.4/8.9	44.7/1.1/3.9
E_g_ (eV) (Tauc)	1.56	1.53	1.53	1.53	1.53	1.53
R_Vis_/R_NIR_/*R*	22/67/44	20/71/46	20/71/46	19/70/45	19/69/44	20/69/45

## Data Availability

Data sharing is not applicable to this article.

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
