# Peer review of "High NIR Reflectance and Photocatalytic Ceramic Pigments Based on M-Doped Clinobisvanite BiVO4 (M = Ca, Cr) from Gels"

_materials, 2023, doi:10.3390/ma16103722_

Round 1

Reviewer 1 Report

There are some observations regarding the weak/incorrect discussion of some data:

- The XRD patterns of the Bi1-xCrxVO4 samples. must include the assignment of the main peaks

- There is no discussion of the thermal analysis data from the “Figure 9. DTA and TG of employed mineralizers.”

- The peaks observed UV-Vis-NIR spectra Figs 4, 11, 14 needs proper assignment according to the energy levels diagram of the Cr3+ ions (d3 configuration) in the local crystalline symmetry and a reference. The 300nm peak needs also an assignment, its magnitude is not dependent on the Cr concentration.

- In the Figure 5. must be included the spectrum for undoped sample and discussed by comparison to the doped samples.

- In the Figure 6. Tauc plots is applicable/useful only for undoped sample…for the other ones the plots show only the peak at around 600nm (about 2eV) and therefore the band gap analysis is distorted and incorrect.

- An absorption or emission band is due to a transition and NOT to the glaze or turquoise as the authors claim in the discussion of the Figs. 5 and 6.

Besides this:

- The authors have to include in the introduction a section/few statements with the questions and the main aim of their study and in the conclusion a section with the advance and the importance of the new knowledge acquired compared to the previous studies

Author Response

Attached you will find our response to your kind comments and suggestions. Thank you so much.

Reviewer 2 Report

Seem:

a) The work has scientific relevance and merit, and is very interesting for the ceramic pigment industry.

b) Abstract must be modified for not meeting the objectives of the work in relation to the conclusions.

c) The introduction and review of the literature partially meet the objectives of the work, however, it requires a review of the literature of the last 05 years.

d) Materials and Methods: Methodology is well written in relation to the experimental part.

e) Discussion :

Figures 1, 2, 3, 4, 5, and 6 need further discussion in relation to works published in the last 05 years.

Figures 7, 8, 9, 10, 11, 12, 13, 14, 15, 16, 17, 18, 19, and 20 and Tables 3 and 4 were not discussed and compared with literature data in the last 05 years.

f) The reviewer suggests that the discussion be redone based on the literature of the last 05 years.

g) Conclusion - they do not meet the objectives of the work.

h) This reviewer understands that the work must be published after major modifications.

Quality of English Language very good.

Author Response

(The authors gave the same response as above.)

Reviewer 3 Report

Preparation from gels and investigations on pigments based on Ca, Cr – doped BiVO4 are reported in this manuscript. The authors report a continuous increase of the NIR reflectance with calcium content. However, this increase is not larger than 9% when the calcium content is increased from zero (pure BiVO4) to 0.4, which corresponds to (Ca0.4 Bi0.6)VO4. The manuscript contains an original work. However, I cannot recommend the publishing of this paper before the authors modify the manuscript to eliminate any suspicion of plagiarism or self plagiarism. I will refer to two resources:

[Ref A]  a paper published before by the same authors Guillermo Monrós *, José A. Badenes and Mario Llusar, “Ecofriendly High NIR Reflectance Ceramic Pigments Based on Rare Earths Compared with Classical Chromophores Prepared by DPC Method”, Ceramics MDPI 2022, 5, 614–641. https://doi.org/10.3390/ceramics5040046

[Ref B] Internet resource, http://www.huevaluechroma.com/011.php  , David Briggs, “the Dimensions of Colour”,

1. At lines 98-112 in the present manuscript, the text is identical with text from Introduction of the [Ref A – pg. 614-615].

The CIEL*C*h color space (similar to CIEL*a*b*) correlates well with the color perception of the human eye. It has the same diagram as the L*a*b* color space but uses cylindrical coordinates instead of rectangular coordinates. In this color space, L* indicates lightness like in CIEL*a*b* model, C* represents chroma, and h* is the hue angle. The value of chroma C* is the distance from the lightness axis (L*) and starts at 0 in the center. Hue angle starts at the +a* axis and is expressed in degrees (e.g. 0° is +a* axis, or red, and 90° is +b axis, or yellow). The values of C* and h* can be estimated from a* and b* parameters by the equations 1 and 2 respectively: C* = (a2 + b2 )^1/2 (1) h* = arctan (b*/a*) (2) Color systems based on hue, lightness and relative chroma first appeared in the early 19th century, but the key concept of absolute chroma was devised by the American artist Albert Munsell.

2. At lines 110-112 in the present manuscript, the text is identical with some text from ][Ref B].

“ Color systems based on hue, lightness and relative chroma first appeared in the early 110 19th century, but the key concept of absolute chroma was devised by the American artist 111 Albert Munsell.”

3. At lines 118 – 119 in the present manuscript, the text is identical with some text from [Ref A – Pg. 615].

“Chroma or color strength refers to 118 the amount of visual difference from a grey of the same value.”

4. At lines 112 – 119 in the present manuscript, the text is identical with some text from [Ref B].

“ … classification of colors of objects according to dimensions of hue, lightness (or greyscale value) and chroma (or relative chroma, often loosely referred to as "saturation"). Hue refers to the circular scale of "pure" or "saturated" colors formed by the colors seen in the spectrum (red, orange, yellow, green, cyan, blue and violet), together with the non-spectral colors like magenta, seen when the two ends of the spectrum are mixed. Lightness refers to the scale from black to white; tone, value and greyscale value are synonyms or very closely related. Chroma or color strength refers to the amount of visual difference from a grey of the same value.”

5. Please correct (CrxBi1-x)VO in Table 4

6. The authors are kindly asked to carefully rewrite the text. The manuscript contains such non intelligible parts as the last paragraph in the Conclusions (lines 481-484):

“The Cr doped samples an interesting degradation half time of 85 min that increases to 150 min the citrate sample but much inferior to simple photolysis of the Orange II of the CONTROL test due probably to the high agglomeration of the fine particles of BiVO4 MOD samples observed by SEM microscopy.”

The manuscript contains such non intelligible parts as the last paragraph in the Conclusions (lines 481-484):

“The Cr doped samples an interesting degradation half time of 85 min that increases to 150 min the citrate sample but much inferior to simple photolysis of the Orange II of the CONTROL test due probably to the high agglomeration of the fine particles of BiVO4 MOD samples observed by SEM microscopy.”

Author Response

(The authors gave the same response as above.)

Round 2

Reviewer 1 Report

The authors provided the proper answers to the comments raised.

Author Response

Thank you very much for your kind comments and suggestions.

Reviewer 2 Report

All corrections suggested by this reviewer have been implemented. This reviewer understands what the article may be published.

Author Response

(The authors gave the same response as above.)

Reviewer 3 Report

At line 85

instead “Fatwa et al. (10)”

should be “Fatwa et al. [10]”

At line 86

instead “calcium as an n dopant into BiVO4”

should “be calcium as an acceptor type dopant into BiVO4”

Please add in Fig.5 also the spectrum for the BiVO4 sample, i.e.   without Cr doping

I strongly encourage the authors to add the DOI to all the cited journal papers in the References list.

Author Response

Thank you very much for your kind comments and suggestions. We have reviewed all the comments and believe that the document has been improved with the reviewer suggestions.

At line 85

instead “Fatwa et al. (10)”

should be “Fatwa et al. [10]”

RESPONSE

Thank you for your kind warning. The error is solved.

At line 86

instead “calcium as an n dopant into BiVO4”

should “be calcium as an acceptor type dopant into BiVO4”

RESPONSE

Thank you for your kind warning. The error is solved.

Please add in Fig.5 also the spectrum for the BiVO4 sample, i.e.   without Cr dòping

RESPONSE

Thank you for your suggestion. The spectrum for without Cr doping has been added at Fig. 5

I strongly encourage the authors to add the DOI to all the cited journal papers in the References list.

RESPONSE

Thank you for your suggestion. The DOI is added to all the cited journal papers in the References.